# Integrated Behavioral and Proteomic Characterization of MPP^+^-Induced Early Neurodegeneration and Parkinsonism in Zebrafish Larvae

**DOI:** 10.3390/ijms26146762

**Published:** 2025-07-15

**Authors:** Adolfo Luis Almeida Maleski, Felipe Assumpção da Cunha e Silva, Marcela Bermudez Echeverry, Carlos Alberto-Silva

**Affiliations:** 1Experimental Morphophysiology Laboratory, Natural and Humanities Sciences Center (CCNH), Universidade Federal do ABC (UFABC), São Bernardo do Campo 09606-070, SP, Brazil; adolfomaleski@gmail.com (A.L.A.M.); fhellcunha@gmail.com (F.A.d.C.e.S.); 2Laboratory of Neuropharmacology and Motor Behavior, Center for Mathematics, Computation, and Cognition (CMCC), Universidade Federal do ABC (UFABC), São Bernardo do Campo 09606-070, SP, Brazil; marcela.echeverry@ufabc.edu.br; 3Neuroscience Laboratory, School of Medicine, Universidad de Santander (UDES), Bucaramanga 680006, Santander, Colombia

**Keywords:** Parkinson’s disease, label-free proteomics, oxidative stress, dopaminergic neurodegeneration, locomotor deficits, mitochondrial dysfunction, neurotoxicology, in vivo screening

## Abstract

Zebrafish (*Danio rerio*) combine accessible behavioral phenotypes with conserved neurochemical pathways and molecular features of vertebrate brain function, positioning them as a powerful model for investigating early neurodegenerative processes and screening neuroprotective strategies. In this context, integrated behavioral and proteomic analyses provide valuable insights into the initial pathophysiological events shared by conditions such as Parkinson’s disease and related disorders—including mitochondrial dysfunction, oxidative stress, and synaptic impairment—which emerge before overt neuronal loss and offer a crucial window to understand disease progression and evaluate therapeutic candidates prior to irreversible damage. To investigate this early window of dysfunction, zebrafish larvae were exposed to 500 μM 1-methyl-4-phenylpyridinium (MPP^+^) from 1 to 5 days post-fertilization and evaluated through integrated behavioral and label-free proteomic analyses. MPP^+^-treated larvae exhibited hypokinesia, characterized by significantly reduced total distance traveled, fewer movement bursts, prolonged immobility, and a near-complete absence of light-evoked responses—mirroring features of early Parkinsonian-like motor dysfunction. Label-free proteomic profiling revealed 40 differentially expressed proteins related to mitochondrial metabolism, redox regulation, proteasomal activity, and synaptic organization. Enrichment analysis indicated broad molecular alterations, including pathways such as mitochondrial translation and vesicle-mediated transport. A focused subset of Parkinsonism-related proteins—such as DJ-1 (PARK7), succinate dehydrogenase (SDHA), and multiple 26S proteasome subunits—exhibited coordinated dysregulation, as visualized through protein–protein interaction mapping. The upregulation of proteasome components and antioxidant proteins suggests an early-stage stress response, while the downregulation of mitochondrial enzymes and synaptic regulators reflects canonical PD-related neurodegeneration. Together, these findings provide a comprehensive functional and molecular characterization of MPP^+^-induced neurotoxicity in zebrafish larvae, supporting its use as a relevant in vivo system to investigate early-stage Parkinson’s disease mechanisms and shared neurodegenerative pathways, as well as for screening candidate therapeutics in a developmentally responsive context.

## 1. Introduction

The zebrafish (*Danio rerio*) represents a prominent model for studying neurodegenerative diseases and identifying neuroprotective compounds. Their genetic, anatomical, and neurochemical similarities to mammals, especially regarding dopaminergic pathways, make them ideally suited for modeling early neuronal dysfunction [1,2,3]. In addition, the small size, optical transparency, and external development of zebrafish embryos enable high-throughput behavioral phenotyping and in vivo compound screening [4,5]. In particular, the embryonic and larval stages of zebrafish offer unique advantages for modeling early neurodegenerative events. At this developmental window, the nervous system is rapidly maturing, and the dopaminergic circuitry is already functional and responsive to neuroactive compounds [2,6,7].

Moreover, zebrafish larvae possess a functional blood–brain barrier [8], a conserved repertoire of neurotransmitter systems, and express orthologs of key genes implicated in human neurological disorders [9]. These features allow for the investigation of molecular signatures relevant to early-stage neurodegeneration in a transparent, externally developing vertebrate. Behaviorally, zebrafish larvae display robust and quantifiable locomotor responses to sensory stimuli, such as light flashes, that are modulated by dopamine signaling [10]. This enables the detection of subtle sensorimotor deficits using automated tracking systems, making zebrafish larvae highly suitable for studying both the pathophysiology and functional consequences of dopaminergic dysfunction at stages when therapeutic interventions may still be effective.

Among the neurotoxins commonly used to model neurotoxicity, 1-methyl-4-phenylpyridinium (MPP^+^) remains a well-established tool due to its selective accumulation in dopamine transporter (DAT)-expressing neurons and its well-characterized mechanism of mitochondrial disruption [11,12]. Unlike its prodrug MPTP, which requires enzymatic conversion in astrocytes and presents significant risks of accidental exposure and neurotoxicity for laboratory personnel [13], MPP^+^ is non-volatile, chemically stable, and considerably safer to handle [14]. In the zebrafish model, MPP^+^ has been applied to study dopaminergic system impairment, given its ability to inhibit complex I of the mitochondrial electron transport chain, leading to increased oxidative stress, ATP depletion, and neuronal dysfunction [15,16,17]. While MPP^+^ exposure is frequently used to replicate aspects of Parkinsonian pathology, its effects extend beyond a single disease entity. Many neurodegenerative conditions—including Parkinson’s, Alzheimer’s, and Huntington’s diseases—share early molecular disruptions in mitochondrial homeostasis, redox balance, protein degradation, and synaptic function [18,19]. Therefore, MPP^+^ exposure in zebrafish larvae may serve not only to model dopaminergic stress but also to interrogate the broader network of early molecular perturbations that precede overt neurodegeneration.

Given the progressive and self-amplifying nature of neurodegenerative disorders, therapeutic interventions applied at late stages—when neuronal death and synaptic loss are extensive—often yield only symptomatic relief without halting disease progression [20,21]. In Parkinson’s disease, for example, most diagnosed patients have already lost over 50% of their striatal dopaminergic terminals by the time motor symptoms appear, limiting the capacity for functional recovery [22,23]. This has led to growing interest in the concept of a “therapeutic window” in which early pharmacological interventions, targeting upstream molecular disturbances such as mitochondrial dysfunction, oxidative stress, or impaired proteostasis, may delay or prevent irreversible neuronal loss [24].

The development of experimental models that capture these early stages is therefore essential for identifying compounds with disease-modifying potential. Zebrafish larvae, due to their rapid neural development and transparency, allow real-time monitoring of neuronal function and behavior during a period when dopaminergic damage is still emerging and potentially reversible [25,26]. This makes them a powerful system not only for mechanistic studies but also for high-throughput screening of neuroprotective agents aimed at intercepting neurodegeneration before clinical onset.

In this study, we aimed to comprehensively characterize the behavioral and proteomic alterations induced by acute MPP^+^ exposure in zebrafish larvae, focusing on a critical early developmental window (1–5 dpf) during which neural circuits are still forming and remain responsive to external interventions. By integrating high-throughput behavioral profiling with label-free quantitative proteomics, we identified a phenotype characterized by pronounced locomotor deficits and molecular alterations in pathways commonly associated with early neurodegenerative processes. The observed behavioral impairments closely resemble a Parkinsonian-like phenotype, reinforcing the relevance of this model for investigating early functional deficits associated with Parkinson’s disease. In addition, the affected molecular pathways—including mitochondrial dysfunction, oxidative imbalance, disrupted protein homeostasis, and synaptic alterations—underscore the broader applicability of this system for exploring early neurotoxic mechanisms and screening candidate compounds with potential disease-modifying effects.

## 2. Results

### 2.1. Evaluation of Locomotor Behavior Reveals Parkinsonian Motor Features in MPP^+^-Exposed Zebrafish Larvae

To assess the impact of MPP^+^ exposure on larval motor function, we performed locomotor tracking during a 180 s free-swimming assay. The results, summarized in Figure 1A, show that zebrafish larvae exposed to 500 μM MPP^+^ from 24 to 96 hpf exhibited a marked reduction in total distance traveled compared to vehicle-treated controls. While control larvae displayed robust and variable movement patterns, MPP^+^-treated larvae remained predominantly immobile throughout the assay (*p* < 0.001). Mean velocity did not differ significantly between groups, indicating that when movement occurred, MPP^+^ larvae swam at similar speeds as controls. Likewise, distance per bolt was not significantly different. However, the bolt interval was significantly increased in the MPP^+^ group (*p* < 0.001), reflecting longer pauses between successive burst events and a marked reduction in the frequency of high-activity swimming. The activity bar plot reveals that control larvae alternated between inactive (red) and active (green) states, spending a substantial portion of time engaged in movement. In contrast, MPP^+^-treated larvae were almost entirely inactive throughout the experiment. Tracking plots visually illustrate these differences: control larvae (top row) exhibited dynamic exploration of the wells, with complex swimming trajectories with frequent alternation between stationary (gray), normal (green), and abrupt (red) movement. MPP^+^-exposed larvae (bottom row), in contrast, remained nearly immobile, with minimal and restricted paths.

To further investigate stimulus-evoked motor responses, we analyzed larval locomotion during alternating light (yellow-shaded) and dark periods, as shown in Figure 1B. This approach allows for the assessment of visually driven behavior in addition to baseline activity. In the distance and velocity boxplots, control larvae showed robust increases in movement during light stimulation, while MPP^+^-treated larvae remained hypoactive regardless of the lighting condition. However, velocity did not differ significantly between groups under either condition. The number of bolts was sharply reduced in the MPP^+^ group, while bolt duration and bolt distance showed no significant differences. The locomotor profile plot, showing mean distance moved per second over time, further illustrates these findings: control larvae displayed clear peaks of activity corresponding to light-on periods (highlighted in yellow), rapidly increasing their locomotor output in response to each light stimulus, and then adapting. In contrast, MPP^+^-treated larvae showed a flat, minimal activity profile, with no appreciable response to light stimulation. Collectively, these results indicate that MPP^+^ exposure leads to profound suppression of both spontaneous and light-evoked locomotor behavior in zebrafish larvae. This phenotype is marked by near-complete inactivity and a lack of responsiveness to light stimulation responsiveness, while swimming speed and burst characteristics remain largely unaffected when movements do occur.

### 2.2. MPP^+^ Exposure Elicits Broad Proteomic Dysregulation in Zebrafish Larvae

To characterize the global proteomic response to neurotoxic injury, we performed label-free quantitative proteomics (LFQ) using LC-MS/MS. Of the total proteins identified across all samples, 1363 were shared between the MPP^+^ and control groups (Figure 2A). Protein intensities were quantified using MaxQuant v2.7.0.0 (Max-Planck-Institute of Biochemistry), and differential analysis was conducted in Perseus using Welch’s *t*-test with Benjamini–Hochberg FDR correction (adjusted *p* < 0.05, |log_2_FC| > 1). The resulting volcano plot (Figure 2B) highlights proteins significantly upregulated (right) or downregulated (left) in response to MPP^+^ exposure. A Venn diagram (Figure 2C) illustrates the overlap in protein identifications across conditions, indicating a high degree of shared protein detection, while also revealing group-specific proteins potentially modulated by MPP^+^. To explore the physiological relevance of the identified proteome, we performed tissue enrichment analysis using Metascape’s v3.5 integrated ontology framework. The analysis (Figure 2D) revealed an enrichment of proteins expressed in neural, muscular, and metabolic tissues, supporting the appropriateness of the model system for neurodegenerative studies. The complete list of differentially expressed proteins in the MPP⁺ vs. Control comparison is available in Appendix A.

### 2.3. Top Differentially Expressed Proteins Involved in Neurodegeneration-Related Processes

From the full set of differentially expressed proteins, we curated a subset of 40 proteins with known or putative roles in neurodegeneration. Selection was based not only on statistical significance (adjusted *p*-value < 0.05 and |log_2_FC| > 1), but also on their functional involvement in key neurobiological processes. A heatmap (Figure 3A) constructed using GraphPad Prism displays the log_2_FC expression values for these proteins. The data were hierarchically clustered to emphasize expression trends across treatment conditions, revealing distinct regulatory profiles in MPP^+^-exposed larvae.

To contextualize the potential impact of these alterations, subcellular localization was annotated based on UniProtKB entries. Most proteins were localized to cytoplasmic (40%), membrane-associated (32.5%), and nuclear (17.5%) compartments (Figure 3B), suggesting that MPP^+^ disrupts diverse cellular systems. Functional enrichment analysis using Metascape (GO Biological Process) was performed to identify significantly overrepresented pathways. The 20 most enriched terms (Figure 3C) were selected from distinct functional clusters and ranked by adjusted *p*-values (–log_10_). Enriched terms included mitochondrial translation, vesicle trafficking, trans-synaptic signaling, response to oxidative stress, and proteasome-mediated protein catabolism, all processes known to be involved in neuronal degeneration. Table 1 summarizes the protein’s details.

### 2.4. Coordinated Dysregulation of Parkinsonism-Associated Proteins

To assess the disease relevance of our findings, we curated a subset of eight proteins associated with Parkinsonian mechanisms. Selection was based on disease ontology (UniProt), STRING PDMap associations, and literature evidence, in addition to fulfilling the criteria of significant differential expression (adjusted *p* < 0.05; |log_2_FC| > 1). These proteins include mitochondrial enzymes (SDHA, CS), the redox-regulating protein PARK7 (DJ-1), and several subunits of the 26S proteasome (PSMA5, PSMB, PSMC3, PSMC5, PSMD11) (Figure 4).

The protein–protein interaction (PPI) network presented in Figure 4 was generated using the STRING database (confidence score ≥ 0.7), and it encompasses both physical and functional associations in the literature, including experimentally validated interactions, curated database entries, co-expression data, and predicted functional links. While this broad scope provides a comprehensive view of potential molecular interplay, our interpretation emphasized high-confidence interactions with known relevance to neurodegenerative mechanisms, particularly those affecting mitochondrial function, proteostasis, and synaptic regulation. Nodes were manually categorized into functional groups: mitochondrial metabolism, oxidative stress response, and proteasomal degradation, providing a focused molecular dissection of proteins altered by MPP^+^ exposure in zebrafish larvae that are functionally and pathophysiologically relevant to Parkinson’s disease (PD).

The STRING interaction network (Figure 4A) highlights the tight clustering of proteasomal subunits (e.g., PSMB, PSMA5, PSMC3, PSMC5, PSMD11), mitochondrial enzymes (e.g., SDHA, CS), and oxidative stress regulators such as PARK7 (DJ-1). This network architecture suggests the co-deregulation of essential nodes involved in mitochondrial quality control, ATP generation, and protein homeostasis, all of which are central to early dopaminergic vulnerability in PD [27,28,29]. The downregulation of SDHA and citrate synthase, core components of the TCA cycle—reinforces this bioenergetic collapse and supports the model’s construct validity [30]. Importantly, the detection of DJ-1 (PARK7), a redox-sensitive chaperone with genetic links to familial PD, further supports the engagement of endogenous antioxidant defenses that are insufficient to counteract the sustained mitochondrial insult [31].

The consistent alteration of proteasomal subunits points to a compromised protein degradation capacity, a phenomenon broadly observed in both idiopathic and genetic forms of PD. Dysregulation of the ubiquitin–proteasome system (UPS) not only leads to the accumulation of misfolded or oxidatively modified proteins (including α-synuclein), but also impairs the adaptive stress response necessary for neuronal resilience [32]. PSMC3 and PSMD11, components of the ATPase and regulatory caps of the 26S proteasome, respectively, are particularly relevant, as their altered expression has been implicated in impaired substrate unfolding and gate opening for proteolysis in neurodegeneration models [33].

## 3. Discussion

Initially, our behavioral analysis revealed that MPP^+^ exposure in zebrafish larvae induces a distinct motor impairment profile, characterized by reduced locomotor activity and bradykinesia-like features. These motor deficits closely resemble Parkinsonian phenotypes observed in other experimental models, supporting the use of zebrafish larvae to study functional consequences of early dopaminergic disruption [34,35]. This profound hypokinesia is consistent with previous zebrafish studies using PD neurotoxins, such as MPTP or 6-OHDA, in both larval and adult stages [17]. Such toxin-induced motor slowness in zebrafish closely resembles the bradykinesia observed in PD patients, where dopaminergic neuron loss in the nigrostriatal pathway leads to impaired movement initiation and reduced vigor [36,37]. Notably, the locomotor deficits in our MPP^+^ model occurred without obvious developmental malformations or increased mortality, in line with reports that MPP^+^ at similar doses causes selective neurobehavioral impairment without overt systemic toxicity [12].

A hallmark of our model is the loss of the normal response to light changes. Control larvae in our study displayed the expected burst of locomotor activity upon sudden transitions from light to dark, a startle response mediated by dopaminergic signaling [10]. MPP^+^-treated larvae, in contrast, failed to mount this response, remaining nearly immobile during light-flash stimuli. This abolition of stimulus-evoked hyperactivity suggests impairment of sensorimotor processing, likely involving dopamine-modulated circuits. The absence of light-induced responses, frequently observed in models of dopaminergic stress, aligns with behavioral alterations associated with Parkinsonian states in zebrafish and other vertebrates [12].

Previous studies have demonstrated that neurotoxin-induced dopamine depletion impairs the dark-flash response in larval zebrafish [38] and that MPTP exposure leads to a reduction in visual startle behavior [39,40]. The lack of light response in our MPP^+^ larvae reinforces that dopaminergic deficits underlie their sensorimotor impairment, analogous to the poor reactivity to environmental cues seen in PD patients with midbrain dopamine loss [41].

We also observed that MPP^+^-exposed larvae spent most of the time in an inactive state, with significantly fewer locomotor bursts. When movement did occur, swim speed and burst distance were not markedly different from controls, but the frequency of bursts—defined as initiations of movement—was markedly reduced. This behavior suggests difficulty in movement initiation, resembling the akinesia and intermittent “freezing” episodes in PD [42]. Indeed, quantitative analysis of zebrafish locomotor patterns has linked reduced burst frequency and prolonged quiescence to hindbrain and spinal circuit dysfunction associated with dopamine loss [43]. Our findings were in line with these observations: MPP^+^ larvae exhibit infrequent, short-lived bouts of swimming separated by long pauses, an exact analog of the slowness and hesitancy to initiate movement that is characteristic of Parkinsonism [39,43]. This hypokinetic state in zebrafish has strong face validity for PD: toxin-based rodent models (e.g., MPTP-treated mice, rotenone-treated rats) likewise show bradykinesia and prolonged immobility [44], and even invertebrate models develop reduced motility when dopaminergic networks are disrupted [45,46].

The convergence of these behavioral outcomes across species highlights a conserved mechanism underlying movement initiation. In this context, the zebrafish MPP^+^ model emerges as a valuable tool for investigating behavioral and molecular features of Parkinsonism during early development, when functional impairments may still be modulated by therapeutic intervention.

Mechanistically, our zebrafish data align with the known pathophysiology of MPP^+^ and PD. MPP^+^ is selectively taken up by dopamine transporter-expressing neurons in the zebrafish brain, accumulating in dopaminergic cell bodies of the ventral diencephalon (homologous to the mammalian substantia nigra) [2]. This leads to degeneration of those neurons and a sharp reduction in striatal dopamine levels [47]. Together, these points demonstrate the predictive validity of the model.

In parallel with these behavioral effects, the proteomic alterations observed in MPP^+^-treated zebrafish larvae point to mitochondrial dysfunction and oxidative stress as central components of the induced neurotoxic response. These processes are known to play prominent roles not only in Parkinson’s disease but also in other neurodegenerative disorders, reinforcing the biological relevance of this model for studying early-stage neuropathological mechanisms [15,16]. By blocking NADH dehydrogenase, MPP^+^ causes a rapid drop in ATP production and a surge in reactive oxygen species (ROS) generation [48]. Our proteomic data reflect this bioenergetic crisis: several mitochondrial proteins (including components of oxidative phosphorylation and metabolic enzymes) were significantly downregulated in MPP^+^ larvae, indicating impaired electron flow and energy metabolism. These changes align with the cascade observed in dopaminergic neurons exposed to MPP^+^ in other models, where mitochondrial failure triggers excessive ROS, calcium dysregulation, and ultimately cell death [23,49].

Gene ontology enrichment analysis in our dataset highlighted processes such as mitochondrial translation and oxidative stress response, reinforcing that mitochondrial impairment and an oxidative stress response are occurring in the larvae. Notably, mitochondrial dysfunction and oxidative damage are widely recognized hallmarks of PD pathogenesis [50,51], and post-mortem analyses of PD patient brains have documented deficits in complex I activity, decreased ATP, and elevated oxidative damage to lipids, proteins, and DNA in the nigrostriatal system [52].

While whole-larvae proteomic profiling may raise concerns about the dilution of neuron-specific protein changes due to contributions from non-neuronal tissues, this approach remains a widely adopted and validated strategy for early-stage zebrafish studies, particularly during the 1–5 dpf developmental window when the central nervous system constitutes a major proportion of the total tissue mass and peripheral organs are still under maturation [53,54]. At this stage, the dissection of neural structures alone is technically unfeasible due to the small size and anatomical integration of the larval brain, which precludes accurate isolation without extensive tissue loss or contamination [55]. To mitigate these limitations, we performed tissue enrichment and subcellular localization analyses, which confirmed that many of the differentially expressed proteins were associated with neuronal and mitochondrial compartments—structures highly relevant to neurodegenerative processes. This system-level approach enables detection of biologically meaningful neurotoxic signatures, while future studies employing brain-enriched proteomic strategies will further enhance tissue specificity.

Consistent with findings in mammalian systems, we observed activation of endogenous antioxidant defenses in MPP^+^-treated larvae, with oxidative stress triggering compensatory responses, most notably induction of the Nrf2 pathway, consistent with previous reports in cell and animal models [56,57]. Nrf2 is a redox-sensitive transcription factor that, upon sensing ROS, translocates to the nucleus and upregulates a battery of antioxidant and detoxification genes. Our proteomic data revealed upregulation of several Nrf2-regulated gene products, including proteasome subunits and enzymes involved in glutathione-mediated detoxification [58]. This suggests that an Nrf2-driven protective response was initiated in the MPP^+^ larvae, as cells attempted to counteract the surge of ROS. Such Nrf2 activation under oxidative challenge is well documented; for example, studies have shown that Nrf2 can induce the expression of proteasome genes and heme oxygenase 1 as part of the antioxidant response [59].

The upregulation of these stress-response proteins in our data may therefore represent the larvae’s effort to restore redox homeostasis. In summary, the concordance between our zebrafish proteomic changes and the mitochondrial/oxidative stress demonstrates that MPP^+^ intoxication in zebrafish engages the same deleterious pathways—complex I inhibition, energy failure, and oxidative damage—that are implicated in human dopaminergic neurodegeneration [52,60].

Beyond metabolic disturbances, our results highlight significant changes in protein degradation pathways, particularly the ubiquitin–proteasome system (UPS), with multiple proteasome subunits markedly upregulated in MPP^+^-exposed larvae. This dramatic upregulation likely reflects an acute cellular response to increased protein damage or misfolding under oxidative stress. MPP^+^’s inhibition of mitochondrial function would be expected to generate excessive ROS, which oxidatively damage proteins and promote misfolding [59]. Cells often cope by enhancing proteasome activity to clear such damaged proteins. The observed surge in proteasome components in our zebrafish suggests a compensatory boost in proteolytic capacity, potentially driven by the Nrf2 stress response as discussed above.

It is well established that protein homeostasis (proteostasis) is critically challenged in both PD and toxin-based PD models [36]. Misfolded or oxidatively modified proteins tend to accumulate when the UPS is impaired or overwhelmed, contributing to the formation of protein aggregates and neuronal dysfunction [61]. In human PD brains, there is ample evidence of UPS dysfunction: levels of ubiquitin-tagged protein aggregates are elevated, proteasomal enzymatic activities are reduced in the substantia nigra, and proteasome subunits themselves have been detected within Lewy bodies [62].

Our zebrafish data mirror the early-stage UPS response seen in other models, marked by increased proteasome expression, presumably reflecting an early compensatory effort to clear damaged proteins. This parallels findings in acute MPTP mouse models, where an initial increase in proteasome activity is observed in surviving neurons as a stress response, even though chronic toxin exposure or advanced disease ultimately leads to proteasomal impairment [63]. Notably, one of the most upregulated proteins in our dataset was a catalytic subunit of the 20S proteasome core, suggesting an active upregulation of proteasomal throughput.

Over the longer term, sustained mitochondrial stress and ROS can overwhelm or inhibit the UPS [32]. In chronic PD models and patient tissue, proteasome function becomes compromised, as evidenced by the accumulation of ubiquitinated proteins and impaired clearance of aggregation-prone species like α-synuclein [64]. Thus, proteasome upregulation may act as a double-edged sword: it temporarily enhances clearance of damaged proteins, but if oxidative stress continues, the proteasome itself may incur damage or inhibition [43,65]. Importantly, impairment of the UPS is thought to further exacerbate oxidative injury in a feed-forward loop, since undegraded proteins (and particularly protein aggregates) can sequester and incapacitate components of the proteasome [32,39,43].

A key finding from our proteomic analysis is the marked downregulation of multiple proteins involved in synaptic structure and neurotransmission following MPP^+^ exposure. In particular, we detected large decreases in proteins associated with synaptic vesicle trafficking and the SNARE complex. Such a loss of VAMP2 and related vesicle fusion machinery would be expected to severely impair neurotransmitter release [66]. These data indicate that MPP^+^ intoxication compromises the integrity of synaptic connections in zebrafish larvae, likely leading to diminished synaptic transmission in affected neural circuits [67].

Synaptic vulnerability is increasingly recognized as an early and central event in PD and other neurodegenerative disorders [68]. In toxin-based animal models, this principle holds as well. Notably, MPTP-treated mice exhibit marked loss of synaptic markers such as synaptophysin and PSD-95 in the striatum, correlating with the degeneration of dopaminergic terminals and emerging motor deficit [69]. We see a very similar pattern in our zebrafish model: the proteins that maintain synaptic vesicle release and postsynaptic structure are strongly diminished, which likely underlie the behavioral phenotypes observed in our study, including hypoactivity and loss of sensorimotor responsiveness. Our findings therefore support the idea that synaptic impairment is a key mediator of motor dysfunction in the zebrafish MPP^+^ model, just as synaptic loss contributes to motor symptoms in mammalian PD models. This cross-species convergence underscores synaptic network disruption as a reproducible outcome of dopaminergic neurotoxin exposure in zebrafish.

The translational relevance of the zebrafish MPP^+^ model is underscored by its utility in both mechanistic studies and therapeutic screening. Due to their small size and permeability to waterborne compounds, zebrafish larvae enable rapid testing of multiple candidate compounds for their ability to ameliorate PD-like phenotypes [25]. High-content readouts, ranging from simple locomotor activity to detailed proteomic and imaging endpoints, can be obtained in live larvae, offering a rich platform for drug discovery [70]. A recent high-throughput chemical screen identified several new molecules that improved motor function in Parkinsonian zebrafish, some of which target pathways like oxidative stress and proteostasis [25]. Promising hits from such screens can subsequently be advanced into mammalian models, streamlining the preclinical identification of therapeutic candidates for Parkinson’s disease.

A particularly relevant finding in the context of Parkinson’s disease was the significant upregulation of PARK7 (also known as DJ-1) in MPP^+^-treated zebrafish larvae. DJ-1 is a multifunctional protein with a well-established role in oxidative stress regulation and neuroprotection in PD models [9,51]. In zebrafish, PARK7 expression is induced in response to oxidative challenges, contributing to enhanced cellular resilience and survival [9]. Moreover, DJ-1 modulates the Nrf2 pathway by disrupting its interaction with Keap1, thereby promoting the expression of antioxidant enzymes such as HO-1 and NQO1 in dopaminergic neurons under stress [26]. Thus, the observed upregulation of PARK7 likely represents an endogenous neuroprotective response to MPP^+^-induced oxidative stress, helping to preserve mitochondrial integrity and maintain redox balance. This positions DJ-1 not only as a mechanistic player in PD-related toxicity but also as a valuable proteomic endpoint for the screening of neuroprotective compounds in zebrafish models.

Although MPP^+^ is broadly recognized as a mitochondrial toxin, the zebrafish larvae exposed to this compound exhibit a behavioral phenotype that is closely associated with dopaminergic dysfunction—namely, reduced locomotion and impaired visual-motor responses—both of which are standard readouts in Parkinson’s disease (PD) models using zebrafish [9,25]. This behavioral evidence is supported by proteomic alterations affecting pathways classically implicated in PD, including oxidative stress regulation, proteasome function, mitochondrial metabolism, and synaptic integrity.

Notably, proteins such as PARK7 (DJ-1), VAMP2, and multiple proteasome subunits were found to be differentially expressed, echoing previous findings in rodent MPTP models and human PD brain tissue [19]. While MPP^+^ toxicity may encompass broader neurotoxic responses, the convergence of PD-like functional and molecular signatures in our dataset suggests a selective engagement of Parkinsonian mechanisms.

The integration of behavioral and proteomic data from zebrafish larvae exposed to MPP^+^ enables a comprehensive characterization of early neurotoxic events in a whole-organism in vivo context. Zebrafish possess conserved dopaminergic circuits, mitochondrial regulatory pathways, and neurodevelopmental timelines that closely mirror those of mammals, making them highly suitable for modeling early Parkinsonian pathology [25,71]. The phenotypes observed—including hypolocomotion and altered visual-motor responses—are consistent with early Parkinson’s disease (PD)-like features and align with findings from mammalian MPTP models and human motor symptoms [72,73]. Although larval zebrafish do not fully recapitulate the spatial complexity or chronic progression of PD, the system offers a high-throughput, developmentally responsive platform to investigate conserved neurodegenerative mechanisms and screen candidate therapeutics during critical windows of vulnerability [25,74].

Cross-model comparisons reveal that several differentially expressed proteins identified in our dataset, including PARK7 (DJ-1), VAMP2, and proteasomal subunits such as PSMA1 and PSMC6, are also commonly deregulated in rodent MPTP models and post-mortem brain tissue from Parkinson’s disease patients [34,75,76]. These proteins are centrally involved in key pathogenic pathways such as oxidative stress regulation, mitochondrial metabolism, synaptic vesicle trafficking, and proteostasis, reinforcing the translational consistency of our zebrafish model [27]. Although quantitative comparisons across species are limited by developmental timing and anatomical complexity, the qualitative convergence in core functional categories supports the notion that early-stage molecular disruptions in zebrafish larvae can reflect conserved neurodegenerative mechanisms [71].

Altogether, our results validate the use of zebrafish larvae exposed to MPP^+^ during early development as a relevant and integrative in vivo system for studying the initial molecular disruptions and behavioral phenotypes characteristic of Parkinson’s disease and related neurodegenerative conditions.

Moreover, the proteomic signature of MPP^+^ exposure (mitochondrial deficits, increased proteasome, decreased synaptic proteins) could itself serve as a set of molecular endpoints for evaluating therapeutics compounds that reverse the abnormal protein expression changes—for instance, restoring normal levels of mitochondrial enzymes or synaptic proteins—would represent strong candidates for preclinical development [27]. This system-level approach, combining behavioral assays with proteomic profiling, is a powerful aspect of the zebrafish model that can complement traditional mammalian studies.

The observed locomotor impairments are accompanied by proteomic alterations—including mitochondrial dysfunction, oxidative stress, proteostasis disruption, and synaptic loss—that closely resemble hallmark features of Parkinsonism in mammalian models and human post-mortem tissue. Although the model does not fully reproduce the chronic progression of the disease, the consistent overlap between functional deficits and conserved molecular signatures reinforces its translational value.

Notably, this approach allows for real-time, in vivo assessment of neurodegenerative processes both during their initial stages—when therapeutic intervention may still be effective—and after the onset of behavioral impairments, enabling the evaluation of compounds that may reverse established phenotypes despite ongoing or irreversible damage. By integrating behavioral phenotyping with proteomic profiling, the model provides a robust platform for identifying and testing pharmacological agents capable of either preventing early pathogenic events or attenuating functional deficits associated with more advanced stages of neurodegeneration.

Despite the robustness of the proteomic and behavioral data presented, validation through complementary molecular approaches remains a necessary step to confirm and refine these findings. Differentially expressed proteins such as PARK7 (DJ-1), VAMP2, and proteasome subunits are well-documented in the Parkinson’s disease literature, and their identification reinforces the biological plausibility of the observed alterations. Future studies incorporating orthogonal methodologies, including qPCR and Western blotting, will be essential to elucidate the mechanistic relevance of these proteins and strengthen the translational framework derived from this model system.

## 4. Materials and Methods

### 4.1. Zebrafish Maintenance and Embryo Collection

Adult wild-type zebrafish (*Danio rerio*) were housed in the zebrafish facility of the Experimental Morphology Laboratory at the Federal University of ABC (UFABC) under controlled environmental conditions, with the temperature maintained at 28 °C and a 14 h light/10 h dark photoperiod. Fish were kept in glass tanks filled with distilled water supplemented with sodium chloride (60 μg/mL) and adjusted to pH 7.0. All procedures involving animals complied with the ethical guidelines established by the European Directive 2010/63/EU [77] and the Brazilian National Council for the Control of Animal Experimentation (CONCEA) [78].

Fish were fed twice daily with commercial dry feed, and *Artemia* nauplii were offered the day before breeding to stimulate spawning. For embryo collection, breeding tanks were set up the night before with a mesh separator to prevent egg predation. Fertilized embryos were collected the following morning and transferred to Petri dishes containing E3 embryo medium (5 mM NaCl, 0.17 mM KCl, 0.33 mM CaCl_2_, 0.33 mM MgSO_4_), and incubated at 28 °C until the start of the experimental procedures.

### 4.2. Early Neurotoxic Exposure Protocol Using MPP^+^

Zebrafish larvae exhibiting normal morphology and no detectable malformations were exposed to 1-methyl-4-phenylpyridinium iodide (MPP+; Sigma-Aldrich, St. Louis, MO, USA; Cat. No. D048) at a final concentration of 500 μM, diluted in E3 embryo medium (5 mM NaCl, 0.17 mM KCl, 0.33 mM CaCl_2_, 0.33 mM MgSO_4_; all from Sigma-Aldrich: NaCl, Cat. No. S7653; KCl, Cat. No. P9333; CaCl_2_, Cat. No. C1016; MgSO_4_, Cat. No. M2643). The selection of 500 µM MPP^+^ was based on previous zebrafish studies demonstrating consistent dopaminergic neurotoxicity, including mitochondrial dysfunction and behavioral alterations, without inducing unspecific lethality or major developmental malformations [12,25]. Exposure was carried out from 1 to 5 days post-fertilization (dpf), due to high neurodevelopmental relevance, when the zebrafish central nervous system undergoes rapid maturation, dopaminergic circuits are functionally established, and the blood–brain barrier remains permeable to neurotoxins, ensuring direct central nervous system exposure [12].

### 4.3. Behavioral Analysis

Locomotor activity was quantitatively assessed in zebrafish larvae at 120 h post-fertilization (hpf) as a functional readout for potential neurobehavioral alterations resulting from MPP^+^ exposure under alternating light/dark conditions. Larvae from each experimental group (MPP^+^-exposed or control; *n* = 12 per group) were individually allocated into 24-well plates (one larva per well) containing 2 mL of fresh E3 embryo medium. The plates were placed in a custom-designed behavioral tracking chamber under controlled environmental conditions. Following a 15 min acclimation period in darkness, larval behavior was recorded over 180 s under alternating light/dark cycles (30 s of illumination followed by 30 s of darkness, repeated three times). Video acquisition was performed using a high-resolution digital camera system. Locomotor parameters—including total distance traveled and displacement patterns—were quantified using the Fiji software platform (ImageJ; NIH, Bethesda, MD, USA) with established tracking plugins [79,80].

### 4.4. Sample Preparation for Proteomic Analysis

Proteomic samples were obtained in duplicate (*n* = 30 larvae per replicate) from both control and MPP^+^-exposed groups. Larval pools were dried using a SpeedVac concentrator (Thermo Scientific Savant SPD120; Thermo Fisher Scientific, Waltham, MA, USA) and resuspended in 100 μL of extraction buffer containing 150 mM Tris-HCl (pH 8.8), 8 M urea (Sigma-Aldrich, St. Louis, MO, USA; Cat. No. U5378), and 0.5% 1-S-octyl-β-D-thioglucopyranoside (OG; Pierce, Thermo Fisher Scientific, Rockford, IL, USA; Cat. No. 28351). Samples were subjected to three cycles of ultrasonication (1 min each at 30% amplitude, with 2 s rest between cycles) while maintained on ice. After sonication, lysates were centrifuged at 10,000× *g* for 10 min at 4 °C, and the supernatant was collected. Protein precipitation was performed by adding 800 μL of acetone:ethanol:formic acid (50:49.5:0.5, *v*/*v*/*v*) and incubating at −20 °C for 3 h. Precipitated proteins were recovered by centrifugation at 12,000× *g* for 10 min at 4 °C, and the resulting pellet was resuspended in 50 μL of Tris-HCl buffer (150 mM, pH 8.8).

### 4.5. Protein Quantification

Protein concentration was determined using the Bradford assay with the Protein Assay Dye Reagent Concentrate (Bio-Rad, Hercules, CA, USA; Cat. No. 500-0006). A standard calibration curve was generated using serial dilutions of bovine serum albumin (BSA; 200 mg/mL, Sigma-Aldrich, St. Louis, MO, USA; Cat. No. P5369-10 mL). Sample aliquots were diluted 1:20 in deionized water, and 20 μL of each sample was loaded in duplicate into a 96-well microplate. Absorbance at 595 nm was measured using a microplate spectrophotometer (SpectraMax Plus 384, Molecular Devices, San Jose, CA, USA), and protein concentrations were calculated based on the BSA standard curve.

### 4.6. Mass Spectrometry Analysis

Proteomic analysis was performed on an Orbitrap Eclipse Tribrid mass spectrometer (Thermo Fisher Scientific, Bremen, Germany) coupled to a Dionex Ultimate 3000 RLSCnano system (Thermo Fisher Scientific, Germering, Germany). Approximately 1 μg of each digested peptide sample was injected onto a NanoEase M/Z Peptide BEH C18 analytical column (Waters Corporation, Milford, MA, USA) (130 Å, 1.7 μm, 75 μm × 250 mm; Waters) at a flow rate of 300 nL/min. Peptides were separated using a 90 min linear gradient from 4% to 50% acetonitrile ((LC-MS grade); Sigma-Aldrich, St. Louis, MO, USA) in 0.1% formic acid. Full MS scans were acquired over an *m*/*z* range of 375–1500 at a resolution of 120,000, with an AGC target of 1 × 10^6^ and a maximum injection time of 100 ms. The most intense precursor ions were selected for fragmentation via higher-energy collisional dissociation (HCD) at a normalized collision energy of 30%, using an isolation window of 1.2 *m*/*z*, an AGC target of 1 × 10^5^, and an MS/MS resolution of 15,000. Raw data files (*.raw) were converted to mzXML format and analyzed using the Comet search engine (v2018). Database searches were performed against the *Danio rerio* UniProt reference proteome. Peptide-spectrum matches (PSMs) were validated using the PeptideProphet algorithm, with a false discovery rate (FDR) threshold set at ≤3%. Quantification was conducted using the Xpress algorithm. Peptide intensities were aggregated at the protein level using a custom R script (R version 4.3.1; R Foundation for Statistical Computing, Vienna, Austria) to generate relative protein abundance values.

### 4.7. Proteomic Data Analysis and Bioinformatics

Bioinformatic and statistical analyses of proteomic data were conducted using a combination of open-source and web-based tools. To ensure the reliability of differential expression analysis, only proteins consistently detected in both biological replicates of the Control and MPP^+^ groups were retained. Proteins with missing values in either replicate of a group were excluded from statistical comparisons, avoiding imputation strategies and minimizing false positives. This conservative approach prioritized high-confidence quantifications and was suitable for the discovery-oriented nature of the study using whole-larva samples at early developmental stages. Protein–protein interaction (PPI) networks were constructed using the STRING database (v11.5; https://string-db.org; accessed on 12 May 2025), with *Danio rerio* as the reference background and a minimum interaction score threshold of 0.4 (“medium confidence”). Only experimentally validated and curated database-derived interactions were included. Functional enrichment analyses, including Gene Ontology (GO) terms and KEGG pathway annotations, were performed in Metascape (https://metascape.org; accessed on 14 May 2025) using default parameters for *Danio rerio*, with a significance cutoff of <0.01 after Benjamini–Hochberg correction for multiple comparisons. PPI network visualization and functional module detection were carried out in Cytoscape (v3.9.1), using the MCODE plugin (version 2.0.0, Cytoscape software) with default settings. Only proteins with a PeptideProphet score corresponding to a false discovery rate (FDR) ≤3% were included in the quantitative analysis. Protein abundance values were log_2_-transformed and normalized prior to clustering and group-wise comparisons.

### 4.8. Statistical Analysis

Statistical analyses were performed using GraphPad Prism version 8.0 (GraphPad Software, La Jolla, CA, USA). Data are expressed as mean ± standard error of the mean (SEM). Unpaired two-tailed Student’s *t*-tests were used for comparisons between two groups. For analyses involving two independent variables (e.g., light/dark cycles), two-way analysis of variance (ANOVA) was applied, followed by Tukey’s post hoc test for multiple comparisons when appropriate. A *p*-value < 0.05 was considered statistically significant.

## 5. Conclusions

This study provides a comprehensive functional and molecular characterization of MPP^+^-induced neurotoxicity in zebrafish larvae, integrating behavioral and proteomic analyses to model early hallmarks of neurodegenerative conditions such as parkinsonism. Building on an established neurotoxic paradigm, the combined evaluation of motor phenotypes and underlying molecular disturbances offers an integrative view of disease-relevant processes. This approach enables the identification of early dysregulated pathways, including those involved in mitochondrial function, proteostasis, and synaptic integrity, while simultaneously capturing overt behavioral impairments.

Accordingly, it establishes a valuable platform for testing candidate compounds not only for their ability to mitigate established functional deficits, but also for their potential to intercept the initial molecular events that precede irreversible neurodegeneration. Although future studies incorporating orthogonal validation techniques are warranted to refine mechanistic interpretations and enhance translational relevance, the findings support the zebrafish model as a powerful tool for exploring disease mechanisms and advancing therapeutic strategies across a broader spectrum of progressive neurological disorders.

## Figures and Tables

**Figure 1 ijms-26-06762-f001:**
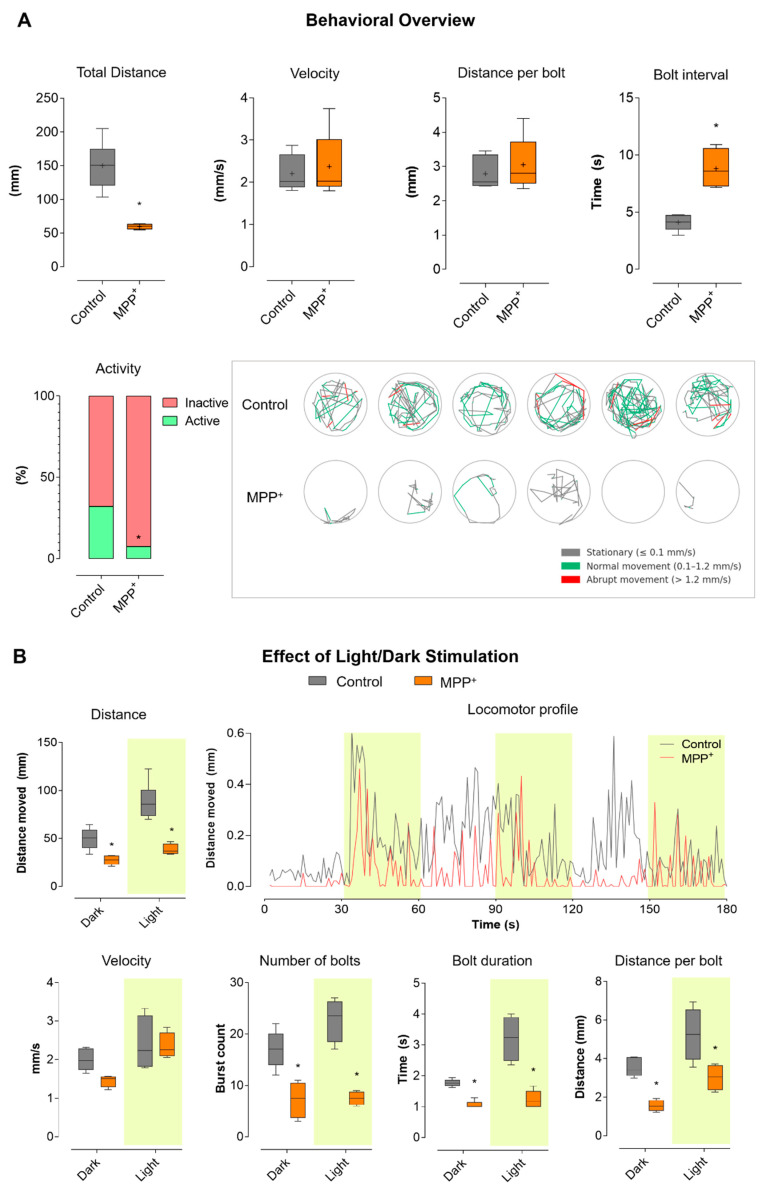
**MPP^+^ exposure induces a Parkinsonian-like behavioral phenotype in zebrafish larvae.** (**A**) Global locomotor parameters: Boxplots display total distance traveled, mean velocity, distance per bolt, and bolt interval for each experimental group (Control, MPP^+^). The “Activity” bar graph shows the proportion of time spent in active (green, velocity > 0.1 mm/s) and inactive (red, velocity ≤ 0.1 mm/s) states. Representative tracking plots illustrate the swimming paths of individual larvae (one per well), with movement phases color-coded as follows: grey for stationary, green for normal movement (0.1–1.2 mm/s), and red for abrupt movement (>1.2 mm/s). (**B**) Light and dark cycle analysis: Behavioral parameters are compared between light and dark periods (highlighted in yellow). Boxplots present total distance, mean velocity, number of bolts, bolt duration, and bolt distance under each condition. The locomotor profile plot displays the average distance moved per second for each group throughout the experiment, with light phases highlighted in yellow. Recordings were obtained using a custom-made behavioral apparatus. Videos were analyzed with Fiji/ImageJ 2, and extracted data were processed and graphed using GraphPad Prism 8. Data are presented as mean ± SEM. Statistical significance was determined by two-way ANOVA followed by Tukey’s post hoc test (*p* < 0.05) and indicated by (*).

**Figure 2 ijms-26-06762-f002:**
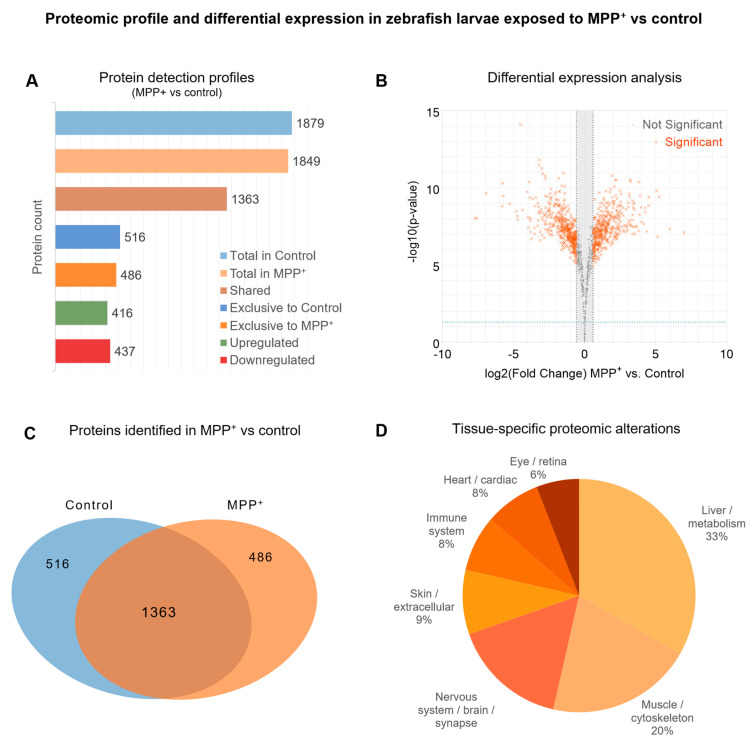
**Overview of proteomic profiling and protein identification metrics following MPP^+^ exposure.** (**A**) Bar chart showing the number of proteins identified in each group (Control and MPP^+^), including shared proteins and those exclusively detected in each condition. (**B**) Volcano plot representing the differential protein expression profile, where significance was determined by applying a threshold of |log_2_ fold change (log_2_FC)| > 1 and adjusted *p*-value (FDR) < 0.05. Proteins above the significance threshold are highlighted. (**C**) Venn diagram indicating the overlap between protein identifications across experimental groups. (**D**) Functional enrichment of identified proteins across major tissue categories based on GO annotations via Metascape. All proteins were quantified using a label-free quantification (LFQ) workflow based on LC-MS/MS.

**Figure 3 ijms-26-06762-f003:**
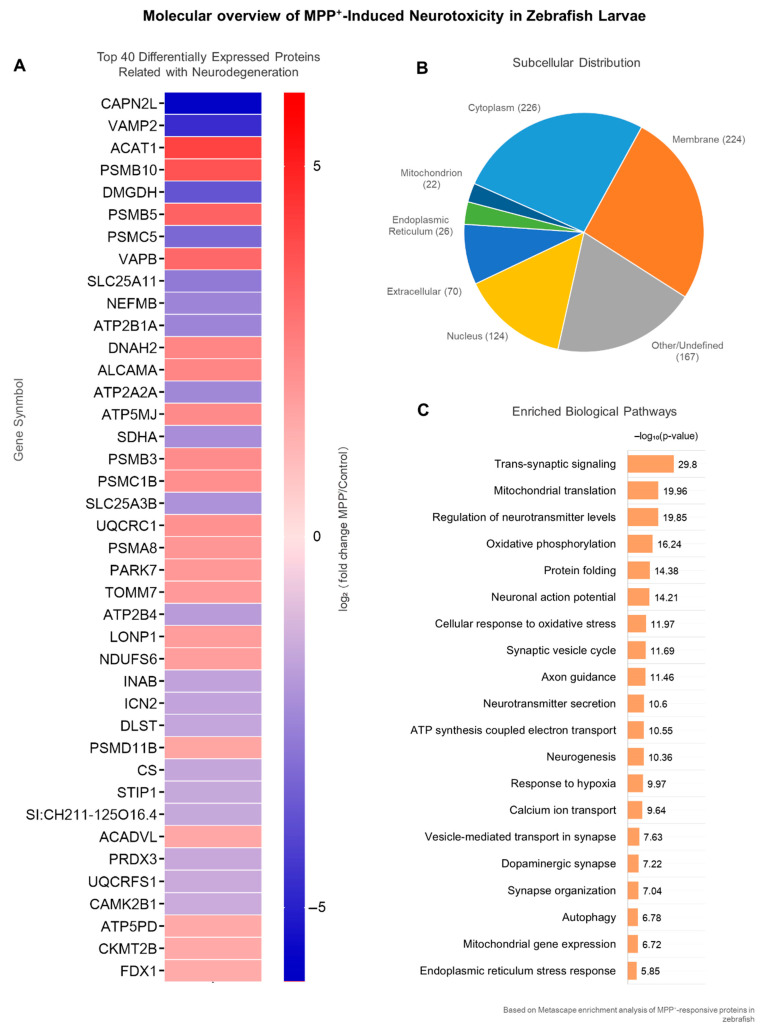
**Top 40 altered proteins involved in synaptic, mitochondrial, and stress-related pathways.** (**A**) Heatmap displaying the top 40 differentially expressed proteins, selected based on adjusted *p*-value < 0.05 and ranked by magnitude of log_2_ fold change (|log_2_FC|). Gene symbols (uppercase, non-italicized) were used for protein labeling in the heatmap for improved readability. Proteins were quantified via LFQ and visualized using hierarchical clustering. (**B**) Subcellular distribution of these proteins, classified according to UniProtKB annotations (accessed in May 2025). (**C**) Biological process enrichment analysis performed using Metascape with GO-BP annotations, employing a hypergeometric test and Benjamini–Hochberg correction to control for multiple comparisons. The top 20 non-redundant terms from distinct functional clusters are shown, ranked by –log_10_ adjusted *p*-values. This analysis highlights functional shifts in mitochondrial translation, trans-synaptic signaling, oxidative stress regulation, and vesicular trafficking reorganization, mechanisms broadly implicated in neurodegenerative diseases, including Parkinson’s and Alzheimer’s. A more detailed overview of the data is provided in Table 1.

**Figure 4 ijms-26-06762-f004:**
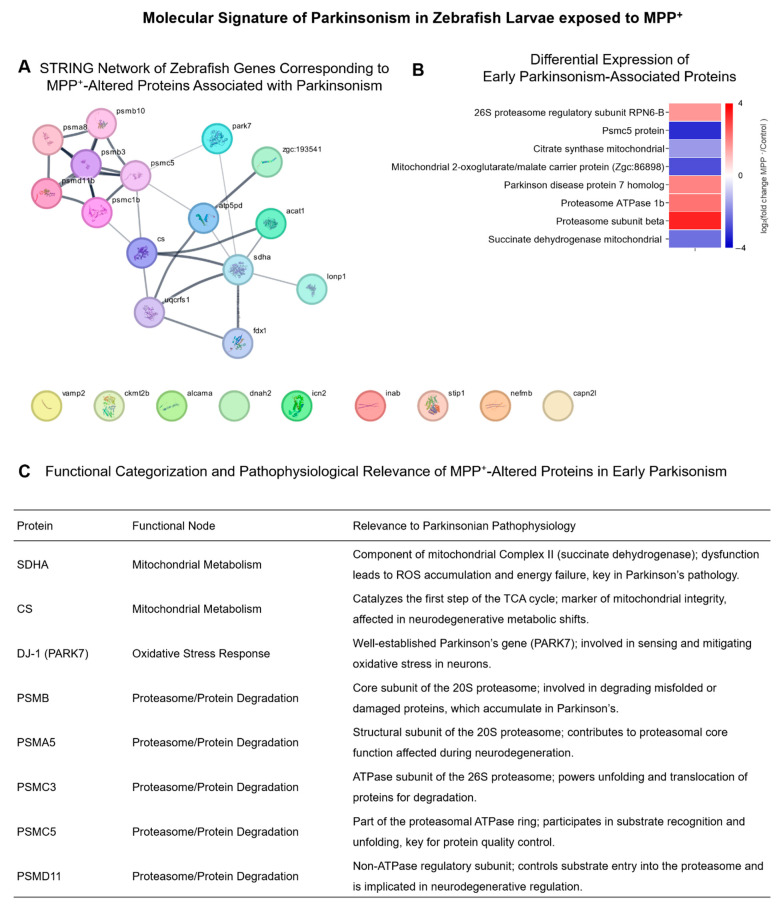
**Parkinsonism-associated proteins exhibit coordinated alterations in key neurodegenerative pathways.** (**A**) Protein–protein interaction network created using STRING (minimum required interaction score ≥ 0.7), representing zebrafish orthologs of eight proteins linked to Parkinson’s disease mechanisms in the literature. Edges represent both experimentally validated physical interactions and predicted functional associations, including curated databases, co-expression data, and computational text mining, as defined by STRING. Proteins were selected based on their biological relevance (involvement in mitochondrial dysfunction, oxidative stress response, or proteasome activity) and significant differential expression in the dataset (adjusted *p*-value < 0.05; |log_2_FC| > 1). Nodes are colored according to manually curated functional categories. (**B**) Heatmap shows the log_2_FC values of these proteins (MPP^+^/Control) based on LFQ intensity values. This targeted subset reflects coordinated molecular disruption in neurodegeneration-relevant processes and validates the suitability of the MPP^+^ model for mimicking Parkinsonian molecular signatures. (**C**) Functional categorization and pathophysiological relevance of the MPP^+^-altered proteins presented in panels (**A**,**B**).

**Table 1 ijms-26-06762-t001:** Differentially expressed proteins related to neurodegeneration in zebrafish larvae exposed to MPP^+^.

Gene Symbol	Protein Name	Log_2_FC	Reg
ACADVL	Very long-chain specific acyl-CoA dehydrogenase, mitochondrial	1.562	↑
ACAT1	Acetyl-CoA acetyltransferase, mitochondrial	4.168	↑
ALCAMA	CD166 antigen homolog A	2.412	↑
ATP2A2A	Calcium-transporting ATPase	−2.398	↓
ATP2B1A	Calcium-transporting ATPase	−2.477	↓
ATP2B4	ATPase, Ca++ transporting, plasma membrane 4	−1.882	↓
ATP5MJ	6.8 kDa mitochondrial proteolipid-like	2.322	↑
ATP5PD	ATP synthase subunit d, mitochondrial	1.499	↑
CAMK2B1	calcium/calmodulin-dependent protein kinase	−1.503	↓
CAPN2L	Calpain-2 catalytic subunit	−5.777	↓
CKMT2B	Creatine kinase S-type, mitochondrial	1.477	↑
CS	Citrate synthase, mitochondrial	−1.614	↓
DLST	Dihydrolipoyllysine-residue succinyltransferase component of 2-oxoglutarate dehydrogenase complex, mitochondrial	−1.652	↓
DMGDH	Dimethylglycine dehydrogenase, mitochondrial	−3.745	↓
DNAH2	Dynein axonemal heavy chain 2 isoform X1	2.464	↑
FDX1	Adrenodoxin, mitochondrial	1.468	↑
ICN2	Protein S100	−1.669	↓
INAB	Internexin neuronal intermediate filament protein, alpha b	−1.694	↓
LONP1	Lon protease homolog, mitochondrial	1.827	↑
NDUFS6	NADH dehydrogenase [ubiquinone] iron-sulfur protein 6, mitochondrial	1.77	↑
NEFMB	Neurofilament medium chain b	−2.489	↓
PARK7	Parkinson disease protein 7 homolog	1.946	↑
PRDX3	Thioredoxin-dependent peroxide reductase, mitochondrial	−1.548	↓
PSMA8	Proteasome subunit alpha type	2.005	↑
PSMB10	Proteasome subunit beta	3.793	↑
PSMB3	Proteasome subunit beta	2.26	↑
PSMB5	Proteasome subunit beta	3.372	↑
PSMC1B	Proteasome	2.198	↑
PSMC5	26S proteasome regulatory subunit 8	−3.234	↓
PSMD11B	26S proteasome non-ATPase regulatory subunit 11B	1.633	↑
SDHA	Succinate dehydrogenase [ubiquinone] flavoprotein subunit, mitochondrial	−2.272	↓
SI:CH211-125O16.4	Neuroblast differentiation-associated protein AHNAK	−1.567	↓
SLC25A11	Mitochondrial 2-oxoglutarate/malate carrier protein	−2.773	↓
SLC25A3B	Solute carrier family 25 member 3	−2.134	↓
STIP1	Stress-induced-phosphoprotein 1	−1.573	↓
TOMM7	Mitochondrial import receptor subunit TOM7 homolog	1.909	↑
UQCRC1	Cytochrome b-c1 complex subunit 1, mitochondrial	2.115	↑
UQCRFS1	Cytochrome b-c1 complex subunit Rieske, mitochondrial	−1.516	↓
VAMP2	Vesicle-associated membrane protein 2	−4.728	↓
VAPB	Vesicle-associated membrane protein-associated protein B/C isoform X1	3.207	↑

Gene symbols, protein names, and log_2_ fold change (log_2_FC) values are shown for proteins differentially expressed in zebrafish larvae exposed to MPP^+^ compared to control. Direction of regulation is indicated by arrows (↑ upregulated, ↓ downregulated). All values shown reached statistical significance (adjusted *p* < 0.05).

## Data Availability

The full proteomic dataset supporting this study is publicly available on Mendeley Data at https://doi.org/10.17632/z2jfvjydh2.1.

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
