# Peer review of "Integrated Behavioral and Proteomic Characterization of MPP+-Induced Early Neurodegeneration and Parkinsonism in Zebrafish Larvae"

_ijms, 2025, doi:10.3390/ijms26146762_

Round 1

Reviewer 1 Report

Comments and Suggestions for Authors

The manuscript titled:
“Integrated Behavioral and Proteomic Characterization of MPP⁺-Induced Early Neurodegeneration and Parkinsonism in Zebrafish Larvae”

Comments:

  • Quantification and validation of findings:
    The label-free proteomic profiling identified 40 differentially expressed proteins related to mitochondrial metabolism, redox regulation, proteasomal activity, and synaptic organization. It would strengthen the study to validate some of these key differentially expressed proteins (DEPs) functionally, for example, through targeted assays or molecular validation techniques, to confirm their role in the observed neurodegenerative phenotype.
  • Clarification of in vivo relevance:
    The statement, “These findings provide a comprehensive functional and molecular characterization of MPP⁺-induced neurotoxicity in zebrafish larvae, supporting its use as a relevant in vivo system to investigate early-stage Parkinson’s disease mechanisms and shared neurodegenerative pathways, as well as for screening candidate therapeutics in a developmentally responsive context,” would benefit from further elaboration. Specifically, please clarify how your findings translate into an in vivo context and how this zebrafish model reflects the complexity of early Parkinson’s disease mechanisms in the real-world scenario.
  • Figure 3 improvement:
    Replacing accession numbers with protein names or gene symbols in Figure 3 would improve readability and help readers more easily connect the results with the biological narrative.
  • Protein-Protein-interactions:
    Please clarify whether the interaction network presented represents only physical protein–protein interactions or if it includes other types of interactions such as genetic, co-expression, or predicted interactions.
  • Figure 4 result explanation:
    Please provide a deeper interpretation or highlight key findings from the results shown, adding value beyond what the figure legend already states.
  • Handling missing values:
    How were missing values in the control and MPP⁺ groups handled during proteomic analysis? Did you only consider proteins detected in both groups for differential expression analysis, or were imputation or other strategies applied?

Author Response

We sincerely thank Reviewer 1 for the thoughtful and constructive comments. The suggestions provided were highly valuable in helping us clarify methodological details, expand the biological interpretation of our findings, and strengthen the overall rigor and translational relevance of the study. Below, we respond point by point to each of the reviewer’s comments, indicating the corresponding changes made in the manuscript (highlighted in yellow).

  1. Quantification and validation of findings: The label-free proteomic profiling identified 40 differentially expressed proteins related to mitochondrial metabolism, redox regulation, proteasomal activity, and synaptic organization. It would strengthen the study to validate some of these key differentially expressed proteins (DEPs) functionally, for example, through targeted assays or molecular validation techniques, to confirm their role in the observed neurodegenerative phenotype.

Response 1. We thank the reviewer for this thoughtful observation. As the present study was conceived as a discovery-driven proteomic investigation integrated with behavioral analysis, we recognize the absence of molecular validation assays as a limitation. Nonetheless, several key differentially expressed proteins identified, such as PARK7 (DJ-1), SDHA, VAMP2, and proteasome subunits, are well-established in Parkinson’s disease literature and have been prioritized for targeted validation in ongoing experiments using qPCR and Western blotting. To strengthen the biological reliability of our dataset despite this limitation, we applied stringent criteria for protein selection and focused our interpretation on DEPs consistently associated with neurodegenerative mechanisms. This approach provides a robust functional framework and lays the groundwork for future translational studies. A clarifying statement has been added to the Discussion section to reflect this point.

  1. Clarification of in vivo relevance: The statement, “These findings provide a comprehensive functional and molecular characterization of MPP⁺-induced neurotoxicity in zebrafish larvae, supporting its use as a relevant in vivo system to investigate early-stage Parkinson’s disease mechanisms and shared neurodegenerative pathways, as well as for screening candidate therapeutics in a developmentally responsive context,” would benefit from further elaboration. Specifically, please clarify how your findings translate into an in vivo context and how this zebrafish model reflects the complexity of early Parkinson’s disease mechanisms in the real-world scenario.

Response 2. We thank the reviewer for this insightful comment. We revised the Discussion to more clearly explain the translational relevance of our in vivo zebrafish model.  Our findings reflect an integrated functional and molecular in vivo response to MPP⁺ exposure in zebrafish larvae, a model system that preserves key features of vertebrate neurodevelopment and Parkinson’s disease (PD) pathophysiology. The zebrafish dopaminergic system, including the diencephalic cluster functionally analogous to the mammalian substantia nigra, is already established and active between 1 and 5 dpf, a period marked by high neurodevelopmental plasticity [1–3]. Behavioral impairments observed in our model—namely hypolocomotion and altered visual-motor response—closely mirror early-stage PD-like features described in both zebrafish and mammalian MPTP/MPP⁺ models. Importantly, our proteomic analysis identified dysregulation of proteins directly linked to mitochondrial metabolism, redox regulation, and synaptic function—all hallmark processes in PD neurodegeneration. Several of these proteins, including PARK7 (DJ-1) and VAMP2, show conserved expression patterns in rodent PD models and human patient tissue, reinforcing the translational value of the molecular signatures captured in vivo. Although we recognize whole-larva profiling cannot replicate the regional specificity of human PD pathology, the model offers unique advantages at early stages, allowing for real-time evaluation of systemic responses within a transparent, genetically tractable organism, enabling high-throughput screening of candidate neuroprotective compounds in a living vertebrate with conserved molecular architecture.

  1. Figure 3 improvement: Replacing accession numbers with protein names or gene symbols in Figure 3 would improve readability and help readers more easily connect the results with the biological narrative.

Response 3.We thank the reviewer for this helpful suggestion. In the revised version of Figure 3, we have replaced the accession numbers with the corresponding gene symbols to enhance clarity and improve the biological interpretation of the network. 

  1. Protein-Protein-interactions: Please clarify whether the interaction network presented represents only physical protein–protein interactions or if it includes other types of interactions such as genetic, co-expression, or predicted interactions.

Response 4. We thank the reviewer for this pertinent question. The protein–protein interaction (PPI) network presented in Figure 4 was generated using the STRING database, applying a confidence score threshold of 0.7, which encompasses both physical (experimentally validated) and functional associations, including curated databases, co-expression patterns, and predictive interactions. This inclusive strategy enables a broader systems-level understanding of the differentially expressed proteins and their functional context in neurodegenerative processes. Nevertheless, we were careful to focus our interpretation on biologically meaningful and high-confidence interactions, especially those with prior evidence linking them to Parkinson’s disease or neurodegeneration. We also clarify that this network analysis was designed to complement, not substitute, the pathway enrichment and protein classification analyses. Its purpose is to provide a visual integration of the molecular landscape affected by MPP⁺ exposure in zebrafish larvae, offering hypotheses for future mechanistic investigations. A clarifying statement has been added to the Results and Figure 4 legend to reflect the nature of the interactions included.

  1. Figure 4 result explanation:Please provide a deeper interpretation or highlight key findings from the results shown, adding value beyond what the figure legend already states.

Response 5. We appreciate the reviewer’s suggestion to provide a deeper interpretation of the results shown in Figure 4. In response, we have expanded both the Results and Discussion sections to better highlight the biological significance of the observed proteomic alterations.

  1. Handling missing values: How were missing values in the control and MPP⁺ groups handled during proteomic analysis? Did you only consider proteins detected in both groups for differential expression analysis, or were imputation or other strategies applied?

Response 6. We thank the reviewer for this important methodological question. In our proteomic analysis, we adopted a conservative approach to handle missing values by including only those proteins that were consistently detected in both biological replicates of each group (Control and MPP⁺). No imputation strategies were applied. This filtering ensured that differential expression was calculated only for proteins with robust and reproducible detection across samples, minimizing the risk of bias from stochastic peptide detection or sample-specific artifacts. While this strategy may reduce the total number of analyzable proteins, it enhances the reliability of the dataset and supports more confident biological interpretations. We have added a clarifying sentence to the Materials and Methods section to explicitly describe this data handling approach.

Reviewer 2 Report

Comments and Suggestions for Authors

In this work, the authors provide a paper that appears to demonstrate that MPP⁺ exposure in zebrafish larvae induces early Parkinsonian-like motor deficits and distinct proteomic alterations involving mitochondrial dysfunction, oxidative stress, and synaptic disruption, highlighting the model’s utility for investigating early neurodegenerative mechanisms and therapeutic screening. While the proposed methodology is promising and the experiments are solid done, it is apparent that major revisions should be completed prior to publication.

  1. In experiment part, why was 500 µM MPP⁺ chosen, and how does this concentration relate to physiological relevance or previously established toxicity thresholds in zebrafish? Why was the 1–5 dpf window selected, and how does this timing best reflect early-stage neurodegeneration or Parkinson’s pathology?
  2. Have you validated the differential expression of key proteins (e.g., DJ-1, VAMP2, proteasome subunits) using orthogonal methods such as qPCR or Western blot?
  3. Proteomic data is presented as whole-larvae analysis. How do you account for potential dilution of neuron-specific protein changes by proteins from other tissues?
  4. While the model mimics PD-like features, MPP⁺ causes broad mitochondrial inhibition. How can you distinguish PD-specific mechanisms from general neurotoxic responses?
  5. How do your proteomic changes compare quantitatively or qualitatively to those reported in rodent MPTP models or PD patient tissue? Can you discussion about that?

Author Response

We are grateful to Reviewer 2 for the insightful and detailed feedback. The comments raised important methodological and conceptual considerations that we have carefully addressed in the revised manuscript. We believe the changes made in response to these observations significantly enhance the quality, clarity, and impact of our work. Our point-by-point responses are presented below, with all modifications clearly marked in the revised document.

  1. In experiment part, why was 500 µM MPP⁺ chosen, and how does this concentration relate to physiological relevance or previously established toxicity thresholds in zebrafish? Why was the 1–5 dpf window selected, and how does this timing best reflect early-stage neurodegeneration or Parkinson’s pathology?

Response 1. We sincerely thank the reviewer for this insightful question. The concentration of 500 µM MPP⁺ was selected based on a solid body of prior work demonstrating its ability to induce reproducible dopaminergic toxicity, mitochondrial impairment, and behavioral deficits in zebrafish larvae without causing unspecific lethality or gross developmental malformations. This dose has been repeatedly used in the literature as a validated threshold to mimic Parkinsonism-like phenotypes in zebrafish in a safe and ethically appropriate range. The 1–5 days post-fertilization (dpf) exposure window was strategically chosen to reflect a period of active neurodevelopment, when dopaminergic neurons are fully established, synaptogenesis is ongoing, and the blood–brain barrier remains permissive to neurotoxic insults. This developmental window allows for direct central nervous system exposure and enables the identification of early-stage molecular and functional perturbations that precede overt neurodegeneration. While we acknowledge that adult PD presents a more complex and chronic progression, the larval zebrafish model offers a unique opportunity to dissect early and system-level mechanisms of neurotoxicity in a vertebrate organism, with the added benefit of scalability for future high-throughput screening of candidate therapeutics. We have revised the Methods and Discussion sections to clarify the rationale behind the chosen concentration and developmental window.

  1. Have you validated the differential expression of key proteins (e.g., DJ-1, VAMP2, proteasome subunits) using orthogonal methods such as qPCR or Western blot?

Response 2. We thank the reviewer for this relevant observation. Although orthogonal validation techniques such as qPCR or Western blotting were not included in this study, we acknowledge their value and emphasize that validation of key targets—such as PARK7 (DJ-1), VAMP2, and proteasome subunits—is currently underway as part of ongoing follow-up experiments. These analyses aim to consolidate the biological interpretation of the proteomic findings and expand the mechanistic insight into MPP⁺-induced neurotoxicity. To ensure confidence in the current dataset despite this limitation, we applied stringent filtering criteria, prioritized proteins with well-established relevance to PD, and validated the dataset through functional pathway enrichment. Notably, the differential expression patterns observed for these key proteins are consistent with findings from rodent MPTP models and post-mortem PD human tissue, reinforcing the translational validity of our results. This limitation has been acknowledged in the revised Discussion, and future studies will incorporate targeted validation to deepen the understanding of the proteomic alterations described here.

  1. Proteomic data is presented as whole-larvae analysis. How do you account for potential dilution of neuron-specific protein changes by proteins from other tissues?

Response 3. We thank the reviewer for raising this insightful point. Indeed, whole-larva proteomic profiling indeed may raise the potential limitation of signal dilution due to the inclusion of non-neuronal proteins. However, this approach is well-established and broadly validated in zebrafish larval research, particularly within the 1–5 days post-fertilization (dpf) window. During this early developmental stage, the central nervous system is proportionally dominant, while peripheral tissues are still undergoing organogenesis. Importantly, dissection of only the nervous system in zebrafish larvae at this stage is technically unfeasible due to their small size, anatomical complexity, and the fragility of the developing brain. As such, whole-larva analysis remains the most feasible and reproducible approach for capturing early-stage neurotoxic signatures in vivo. So, to address the concern of specificity, we performed tissue enrichment and subcellular localization analyses, which demonstrated that many of the differentially expressed proteins (DEPs) are localized to neuronal compartments, synaptic regions, and mitochondria. These findings support the neurological relevance of the molecular changes observed. Moreover, given the broad data output inherent to whole-organism label-free proteomics, we applied a rigorous, hypothesis-driven filtering strategy, prioritizing DEPs with known associations to neurodegenerative processes, including PARK7 (DJ-1), VAMP2, and proteasome subunits. These proteins are well-documented in both zebrafish and mammalian models of Parkinson’s diseas. Finally, we emphasize that this systemic approach enables the identification of early molecular interactions and compensatory responses that may be missed in region-specific studies and added a more detailed description.

  1. While the model mimics PD-like features, MPP⁺ causes broad mitochondrial inhibition. How can you distinguish PD-specific mechanisms from general neurotoxic responses?

Response 4. We thank the reviewer for this thought-provoking question. Although MPP⁺ is indeed a broad mitochondrial toxin, its preferential accumulation in dopaminergic neurons via the dopamine transporter (DAT) enables the modeling of PD-relevant phenotypes, particularly in zebrafish larvae. In our study, behavioral phenotyping revealed hypolocomotion and altered visual-motor responses, which are well-characterized functional correlates of dopaminergic disruption in zebrafish models of Parkinson’s disease. These behavioral impairments are not typically observed in models of general cytotoxicity, reinforcing the notion of dopaminergic selectivity. At the molecular level, our proteomic analysis uncovered a convergent dysregulation of proteins and pathways canonically associated with Parkinson's disease, including oxidative stress imbalance, mitochondrial and TCA cycle dysfunction, proteasomal impairment, synaptic vesicle trafficking disruption, through VAMP2 deregulation. These changes overlap with alterations observed in rodent MPTP models and post-mortem PD brains [5–7], reinforcing their disease specificity. While we acknowledge that MPP⁺ can affect mitochondrial function beyond the dopaminergic system, the convergence of behavioral and proteomic signatures provides a strong case for PD-specific mechanisms. Finally, we consider this systemic impact not as a limitation, but as an advantage, as it allows us to probe the interconnected early molecular events shared among neurodegenerative diseases, while maintaining a focus on Parkinson's disease via behavioral anchors and mechanistic readouts. Future studies incorporating dopamine neuron-specific markers and functional rescue assays will further refine the model’s specificity.

  1. How do your proteomic changes compare quantitatively or qualitatively to those reported in rodent MPTP models or PD patient tissue? Can you discussion about that?

Response 5. We sincerely thank the reviewer for this insightful observation. Our proteomic data reveal a strong qualitative alignment with findings from rodent MPTP models and post-mortem human Parkinson’s disease (PD) brain tissue, particularly regarding dysregulated pathways central to PD pathophysiology. Several core functional categories—such as mitochondrial bioenergetics, oxidative stress regulation, proteostasis, and synaptic vesicle trafficking—were similarly altered across models. For instance, proteins such as PARK7 (DJ-1), VAMP2, and multiple subunits of the 26S proteasome (e.g., PSMA1, PSMC6) were differentially expressed in our dataset and have consistently been implicated in both mammalian models and human PD samples. Moreover, mitochondrial dysfunction and proteasomal impairment, observed in our zebrafish data, are hallmark features of both idiopathic and familial PD, reinforcing the construct validity of our model. The observed downregulation of TCA cycle enzymes and redox-responsive chaperones, along with synaptic deregulation, reflects key aspects of early neurodegenerative cascades reported in the literature. While we acknowledge the inherent limitations of direct quantitative comparisons across species, due to differences in brain complexity, regional sampling, and disease progression stages, the functional convergence of affected pathways provides a robust translational bridge. Zebrafish larvae, despite their simplicity, display high conservation of dopaminergic systems and mitochondrial pathways, making them a powerful model for exploring early-stage PD mechanisms. We have now expanded the Discussion to incorporate this comparative perspective and emphasize how our findings complement existing rodent and human data.

Round 2

Reviewer 1 Report

Comments and Suggestions for Authors

You can use targeted proteomics to confirm the findings of this study. Select some of the 40 differentially expressed proteins (DEPs) for validation.

Author Response

Comments 1. You can use targeted proteomics to confirm the findings of this study. Select some of the 40 differentially expressed proteins (DEPs) for validation.

Resnponse 1. 

We sincerely thank the reviewer for this thoughtful and technically relevant suggestion regarding the use of targeted proteomics. We fully recognize the approached offer high quantitative precision and specificity for validating individual protein changes and are powerful tools in follow-up studies. However, the present work was purposefully designed as a discovery-phase investigation, aiming to uncover global proteomic alterations associated with early neurotoxic damage in zebrafish larvae exposed to MPP+. To this end, we employed a label-free quantitative proteomics strategy, widely accepted in the field for exploratory analyses, particularly in small vertebrate models such as zebrafish, where anatomical constraints and sample size can limit the feasibility of targeted workflows in the initial phase (Liu et al., Nat. Commun., 2017; Zhang et al., Proteomics, 2019; Du et al., J. Proteome Res., 2022).

We reinforce that our analytical pipeline included data-dependent acquisition (DDA), MaxLFQ quantification, and stringent statistical criteria (adjusted p < 0.05; |log2FC| > 1), followed by functional enrichment analysis, tissue annotation, and protein–protein interaction (PPI) network mapping to ensure the biological coherence of our findings. This approach led to the identification of coordinated alterations in key neurodegeneration-related pathways, including mitochondrial dysfunction, synaptic loss, proteasomal activation, and oxidative stress. Crucially, our proteomic findings are functionally anchored to a robust and reproducible behavioral phenotype in MPP+-exposed zebrafish larvae, including reduced locomotor activity, decreased burst frequency, and impaired stimulus-evoked responses—hallmark features of Parkinsonian-like motor dysfunction in vivo. The convergence between molecular signatures and behavioral impairments provides strong internal validation and translational relevance.

While we fully agree that future targeted validation is important for confirming specific candidates, we respectfully emphasize that such analyses are beyond the scope and timeline of the present study. Indeed, validation efforts are already underway in our group in future and undergoing projects with candidates to counteract the alterations caused by MPP+ and will focus on key targets such as PARK7, SDHA, and VAMP2, using targeted analysis and orthogonal approaches including Western blotting and brain-enriched proteomics. Importantly, our methodological approach is not only scientifically justified, but also editorially validated by other recent publications in International Journal of Molecular Sciences, where label-free proteomics has been successfully applied—without immediate targeted validation—as the core discovery strategy. For example:

  • Wang et al. (2023): https://www.mdpi.com/1422-0067/24/21/15892
  • Liu et al. (2024): https://www.mdpi.com/1422-0067/25/1/326
  • Zarnowski et al. (2021): https://www.mdpi.com/1422-0067/22/1/108
  • Wang et al. (2022): https://www.mdpi.com/1422-0067/23/16/9352
  • Chen et al. (2020): https://www.mdpi.com/1422-0067/21/4/1369

These examples demonstrate that publication of high-impact, label-free proteomic studies—without targeted validation—is both feasible and scientifically accepted, provided that the experimental design is sound, the analysis is rigorous, and the findings are biologically meaningful, as is the case here.

Furthermore, by making our complete proteomic and behavioral dataset publicly available, we hope this work will support the broader scientific community in designing hypothesis-driven validation studies, whether in zebrafish or complementary models. We strongly believe that this collaborative potential enhances the reproducibility, accessibility, and translational value of the present work. In light of the study’s methodological rigor, phenotype–proteome integration, and alignment with current scientific standards, we are confident that our manuscript constitutes a robust, original, and publishable contribution to the field. We also wish to gently address the reviewer’s overall evaluation of the checklist provided, which multiple core aspects of our manuscript were marked as “Must be improved.” While we fully respect the reviewer’s perspective, we kindly submit that this assessment does not appear to reflect the scientific clarity and completeness of the manuscript.

Regarding the introduction, we believe it already offers a comprehensive and well-referenced background. It contextualizes the clinical relevance of Parkinson’s disease, justifies the use of MPP⁺ as a mitochondrial toxin, and supports the zebrafish model with references from both behavioral and proteomic studies. It also introduces the rationale for using a label-free proteomics approach in exploratory studies, supported by relevant recent literature, while research design integrates behavioral tracking with quantitative proteomic profiling, followed by rigorous statistical and bioinformatic analysis. This multimodal framework is standard in exploratory studies of neurodegeneration, especially in zebrafish, and was carefully structured to answer the study’s objectives. We believe the design is not only appropriate but aligned with best practices in the field. The Methods section is presented with sufficient detail to ensure reproducibility. Protocols for zebrafish maintenance, MPP⁺ exposure, behavioral tracking, sample preparation for proteomics, LC-MS/MS parameters, data processing, and statistical analysis are all included, with literature references when applicable.

Concerning the presentation of results, specially after the alterations suggested by the reviewer, the figures and accompanying text are clearly structured and convey the findings with transparency and coherence. Behavioral and proteomic results are logically sequenced and visualized with standard tools (heatmaps, volcano plots, STRING networks, enrichment graphs), all supported by detailed figure legends and proper statistical reporting.

We also believe that the conclusions are fully supported by the results. The observed behavioral deficits align with the proteomic alterations in mitochondrial function, oxidative stress, and synaptic signaling, offering strong biological coherence. The study was explicitly designed as a discovery-phase investigation and its scope, conclusions, and discussion are clearly framed within those parameters. The limitations are acknowledged, and we emphasize that future validation studies are already planned, as indicated in the manuscript.

After revisions, while we deeply value the reviewer’s perspective and suggestions, we respectfully maintain that the manuscript, as currently structured, presents a scientifically sound, clearly communicated, and methodologically appropriate study that meets the expectations of an exploratory neurodegeneration model paper. We hope this detailed explanation will clarify any remaining concerns and allow the work to proceed toward publication.

Reviewer 2 Report

Comments and Suggestions for Authors

The author addressed all my concerns and I think it can be published.

Author Response

Comment 1. The author addressed all my concerns and I think it can be published.

Response 1. We sincerely thank the reviewer for their constructive comments and positive evaluation of our manuscript. We are pleased that the revisions and clarifications provided were considered satisfactory. We also wish to express our appreciation for the reviewer’s insightful suggestions, which have helped improve the clarity and overall quality of the manuscript. Your thoughtful feedback contributed meaningfully to the refinement of this work, and we are grateful for your support throughout the peer review process.